
# A Framework for Automatic Calibration of SWMM Considering Input Uncertainty

Xichao Gao [1,2], Zhiyong Yang [1,*], Dawei Han[2], Guoru Huang[3], Qian Zhu[4]

[1]State Key Laboratory of Simulation and Regulation of Water Cycle in River Basin, China Institute of Water Resources and
Hydropower Research, Beijing, 100038, China; 999gaoxichao@163.com (X.G.), wanghao@iwhr.com (H.W.)
[2]Department of Civil Engineering, University of Bristol, BS8 1TR, UK; d.han@bristol.ac.uk (D.W.)
[3]School of Civil Engineering and Transportation, South China University of Technology, Guangzhou, 510640, China;
huanggr@scut.edu.cn
[4]School of Civil Engineering, Southeast University, Nanjing, 211189, China; zhuqian@seu.edu.cn (Q.Z.)

Correspondence to: Zhiyong Yang (yangzy@iwhr.com)

**Abstract.** A new Bayesian framework for automatic calibration of SWMM, which simultaneously considers both parameter uncertainty and input uncertainty, was developed. The framework coupled an optimization algorithm DREAM and the Storm Water Management Model (SWMM) by newly developed API functions used to obtain and adjust parameter values of the model. Besides, a rainfall error model was integrated into the framework to consider systematic rainfall errors. A case study in Guangzhou, China was conducted to demonstrate the use of the framework. The calibration capability of the framework was tested and the impacts of rainfall uncertainty on model parameter estimations and simulated runoff boundaries were identified in the study area. the results show that calibration considering both parameter uncertainty and rainfall uncertainty captures peak flow much better and is more robust in terms of the Nash Sutcliffe index than that only considering parameter uncertainty.

## 1 Introduction

Urbanization alters the hydrological process significantly, resulting in more runoff volume, less concentration time, and thus more severe flooding. To reveal the hydrological characteristics of urban watersheds, a large number of urban hydrological models have been developed in recent decades which are widely used in urban flooding prediction and water resources management (Bisht et al., 2016; Rodriguez et al., 2008; Rubinato et al., 2013). Among these urban hydrological models, the Storm Water Management Model (SWMM) is one of the most popular models used in predicting runoff quantity and quality from urban drainage systems and has been widely used for analysis, design, and planning related to urban water problems throughout the world. SWMM is developed by the United States Environmental Protection Agency (US EPA) and is open-source to users worldwide. It can be used to simulate both single-event and long-term continuous rainfall-runoff processes in catchments containing different types of gray and/or green infrastructures such as storm drains, sanitary sewer systems, Combined Sewer Systems, and rain gardens. Many parameters are used in the SWMM model to represent catchment characteristics and the complex hydrological and hydraulic processes. Theoretically, most of the parameters can be specified through field investigation. Unfortunately, accurate and detailed information about the urban drainage system is usually



unavailable in many cities and the catchment is often simplified significantly in the model considering calculation efficiency. Therefore, some of the model parameters cannot be specified directly and need to be calibrated with an input-output record

of the hydrological response of the catchment.

Manual "trial-and-error" procedure is widely used to calibrate the parameters of the SWMM model, since the model is relatively complicated and does not contain a built-in calibration program (Duan and Gao, 2019; Rosa et al., 2015; Tsihrintzis and Hamid, 1998). However, the manual "trial-and-error" calibration procedure is relatively subjective and very time-consuming. Whether it can obtain the global best-fit parameter values is determined by the experience and expert

judgement of practitioners. To improve calibration efficiency, many automatic calibration procedures have been developed and integrated with the SWMM model with the development of computer technology. In recent years, complex method (Barco et al., 2008), genetic algorithm (GA) (Wan and James, 2002), non-dominated sorting genetic algorithm-II (NSGA-II) (Shinma and Reis, 2011), non-dominated sorting genetic algorithm-III (NSGA-III) (Swathi et al., 2019), dynamically dimensioned search (DDS) (Behrouz et al., 2020) have been integrated with the SWMM model to calibrate the model

parameters automatically and have obtained acceptable results. However, these algorithms eventually provide a specific optimal parameter set and do not consider the impact of uncertainty on parameter estimation and prediction bounds.

Extensive studies identify that equifinality, i.e. different parameter sets may lead to equally acceptable predictions, is widespread in complex hydrological models including SWMM (Her and Chaubey, 2015; Kelleher et al., 2017; Muñoz et al., 2014; Wagner et al., 2018). It implies that the optimal parameter set obtained by traditional calibration methods without

considering the uncertainty may not represent the real catchment correctly. Sun et al. (Sun et al., 2014) incorporated the generalized likelihood uncertainty estimation (GLUE) method in a SWMM model and specified the parameter uncertainty in a highly urbanized sewershed in Syracuse, NY. James et al. (Knighton et al., 2016) evaluated the parameter uncertainty of a SWMM model developed for the Cathedral Run stormwater wetland using the GLUE method and identified the importance of equifinality and uncertainty within the context of stormwater wetland modeling. Ehsan et al. (Raei et al., 2019) developed

a framework for low impact development-best management practices (LID-BMPs) considering parameter uncertainty with fuzzy α-cut technique. Other studies about uncertainty analysis can also be found in Gorgoglione et al. (Gorgoglione et al., 2019), Sharifan et al. (Sharifan et al., 2010), Zhang et al. (Zhang and Li, 2015), and Bellos et al. (Bellos et al., 2017). In general, most of the studies about uncertainty of SWMM modeling focus on parameter uncertainty instead of the uncertainty associated with forcing data and model structure. While SWMM has been successfully used in many studies and the

uncertainty related to the model structure can be considered to be relatively small, the input data, especially precipitation data, used in urban stormwater modeling can be of high uncertainty. For example, the precipitation data measured by tipping bucket rain gauges suffer from large measurement errors, such as splashing losses, evaporation losses, wind losses, and spatial representation. In some extreme situations, the total error can be up to more than 30% (Dotto et al., 2014). Improper handling of input uncertainty in model calibration may yield grossly misleading and biased parameter estimates, which

cannot represent the real catchment (Chen et al. 2018). The bias in parameter estimations can lead to that the SWMM model calibrated using historical record cannot predict future response correctly and thus restricts the application of the SWMM





model in the designing of flooding control facilities if the forcing uncertainty structure varies between historical and future periods (Kavetski et al., 2006; Zhang et al., 2019). Therefore, both parameter uncertainty and input uncertainty should be considered in the calibration of the SWMM model. A new framework for automatic calibration of the SWMM model
considering the two uncertainty sources is therefore proposed in this study.

To settle this problem, PySWMM, a python packaged SWMM developed by Emnet (https://github.com/OpenWaterAnalytics/pyswmm), is expanded by adding some application programming interface (API) functions, which is used to obtain and adjust SWMM parameters directly while the model is running. We then integrate a Bayesian-based optimal algorithm, entitled differential evolution adaptive Metropolis (DREAM), into a modified SWMM
model to develop a new tool, which is designed to calibrate the SWMM model automatically considering both parameter and input uncertainty. DREAM is a novel adaptive Markov chain Monte Carlo (MCMC) algorithm for estimating the posterior probability density function of model parameters based on the Bayesian framework, considering different sources of uncertainty including parameter uncertainty, input uncertainty, and model structural uncertainty. The algorithm is developed by Vrugt et al. (Vrugt et al., 2009a) and is an adaptation of the shuffled complex evolution Metropolis (SCEM-UA) global
optimization algorithm. It runs multiple chains simultaneously for global searching and automatically tunes the scale and orientation of the proposal distribution during the evolution to the posterior distribution (Vrugt et al., 2008). The algorithm shows good efficiency on complex, highly nonlinear, and multimodal target distributions while maintaining detailed balance and ergodicity (Vrugt et al., 2009a).

A case study in Guangzhou, China, is conducted to demonstrate the use of the proposed framework for automatic calibration
of SWMM and to analyze the impact of input uncertainty (mainly referring to rainfall uncertainty in urban rainfall-runoff simulation) on the estimation of parameter values and the boundaries of simulated runoff.

The study is organized as follows: Section 2 describes the proposed calibration framework for the SWMM model developed in this study. Section 3 introduces an application study to verify the developed calibration framework. The results of the application study are given in Section 4, and a short conclusion of the study is presented in Section 5.

## 2 Methods

### 2.1 The SWMM model

SWMM simulates rainfall-runoff process on a collection of sub-catchment areas which receive precipitation and generate runoff and pollutant loads, and transport the generated runoff through a system of pipes, channels, storage/treatment devices, pumps, and regulators (Rossman and Huber, 2016).

In this study, PySWMM, a Python packaged version of SWMM 5.1 (https://github.com/OpenWaterAnalytics/pyswmm), was used instead of the original version of SWMM 5.1. PySWMM is free software developed by Emnet. It provides a Python interface for SWMM. With the PySWMM, users can control the functions and objects in SWMM through Python and develop algorithms exclusively in Python to control the calculation process of SWMM. Some API functions are developed



and added into the model to obtain and adjust parameter values, such as impervious rate and maximum infiltration rate of a
sub-catchment in the SWMM model, while the model is running. With these API functions, the automatic calibration
algorithm and the SWMM model can be connected easily through a Python platform. A rainfall error model used to describe
the rainfall errors, which will be illustrated below in detail, is also integrated into the SWMM model to consider rainfall
uncertainty.

## 2.2 Calibration of the SWMM model

For prediction purpose, the parameter values should accurately reflect the invariant properties of the specific system that they
represent(Vrugt et al., 2008). However, although most of the parameters in the SWMM model have physical meanings, there
are some parameters which cannot be measured directly. Therefore, these parameters need to be meaningfully derived
through calibration against historical records of some state quantities of the catchment.

The hydrological simulation of a catchment by the SWMM model can be expressed by Eq. 1:

$$Y = f(\theta, \hat{\zeta}, \hat{\phi}) \tag{1}$$

where $Y = \{y_1, \dots, y_n\}$ represents the simulated results of the model such as streamflow; $\hat{\zeta}$ refers to the measured boundary
such as precipitation and evapotranspiration; $\hat{\phi}$ represents the initial conditions; $\theta = \{\theta_1, \dots, \theta_d\}$ represents model
parameters; $f$ represents the deterministic or stochastic transition function (the SWMM model in this study). Let $\hat{Y} = \{\hat{y}_1, \dots, \hat{y}_n\}$ represents measurements of observed system behavior (streamflow in this study). The difference between $Y$ and
$\hat{Y}$ can be mathematically expressed by the residual vector (Eq. 2):

$$\varepsilon_i(\theta | \hat{Y}, \hat{\zeta}, \hat{\phi}) = y_i(\theta | \hat{\zeta}, \hat{\phi}) - \hat{y}_i \quad i = 1, \dots, n \tag{2}$$

The calibration is to make the residuals as close as zero. The traditional approach is to build an objective function for the
residuals, such as the sum of squared residuals (which is shown in Eq. 3), and to minimize the objective function through
tuning the values of the parameters.

$$SSR(\theta | \hat{Y}, \hat{\zeta}, \hat{\phi}) = \sum_{i=1}^{n} \varepsilon_i(\theta | \hat{Y}, \hat{\zeta}, \hat{\phi})^2 \tag{3}$$

The approach mentioned above only focuses on errors caused by the model parameters but ignores the errors associated to
model inputs, initial conditions, model structures, and measurements of observed system behavior (streamflow in this paper).
Therefore, it may not obtain the "best-fit" values of the parameters or obtain fake "best-fit" parameters that do not represent
the properties of the real-world hydrologic system. Physically, the errors associated with initial conditions can be eliminated
with the model running. Since the model input, such as precipitation and evapotranspiration, is of high spatio-temporal
heterogeneity, the errors that associated with the model input are much greater than that associated with streamflow
measurements. Consequently, the residual vector between $Y$ and $\hat{Y}$ can be rewritten as Eq. 4:

$$\varepsilon_i(\theta, s(\hat{\zeta}) | \hat{Y}, \hat{\phi}) = y_i(\theta, s(\hat{\zeta}) | \hat{\phi}) - \hat{y}_i \quad i = 1, \dots, n \tag{4}$$

where $s$ represents the error function of the model input.





An algorithm which can consider different sources of error is needed to minimize the residuals expressed in Eq. 4. Bayesian statistics coupled with Monte Carlo sampling is a practical method to settle this problem(Liu and Gupta, 2007; Vrugt et al., 2009b). It can estimate different sources of errors simultaneously and give the posterior distribution of the parameters of the SWMM model and the input errors model. Assuming that the residuals described in Eq. 4 are independent and Gaussian-distributed with constant variance ($\sigma_e$) and a constant mean (0), the posterior probability density function of the parameters,

including parameters of the SWMM model and the input errors model, can be identified as:

$$p\big(\theta, s(\hat{\zeta})|\hat{Y}, \hat{\phi}\big) = c \cdot p(\theta) \cdot p\left(s(\hat{\zeta})\right) \cdot \prod_{i=1}^{n} \frac{1}{\sqrt{2\pi}\sigma_e} exp\left(-\frac{\left(y_i(\theta, s(\hat{\zeta})|\hat{\phi})-\hat{y}_i\right)^2}{2\sigma_e^2}\right) \quad (5)$$

where $c$ is a normalizing contact; $p(\theta)$ and $p\left(s(\hat{\zeta})\right)$ represent the prior distribution of $\theta$ and $s(\hat{\zeta})$, respectively.

As the probability distribution defined in Eq. 5 cannot be derived through analytical analysis, a new developed MCMC method called DREAM was introduced to generate samples from the posterior probability distribution.

**2.3 Input error model**

As described in Section 2.2, input uncertainty should be considered in the model calibration. For the urban rainfall-runoff simulation by the SWMM model, the most important input data is rainfall. Therefore, in this study, only the input uncertainty associated with rainfall is considered.

Storm depth multiplier model (Kavetski et al., 2003) is chosen to consider the uncertainty associated with rainfall forcing.

The model introduces a multiplier for each rainfall event based on the idea that the rainfall depth measurements may have the systematic error for each storm caused by the movement of the storm cell within the catchment but the internal storm pattern may be kept relatively well (Kavetski et al., 2006). Through tuning these multipliers ($m = \{m_1, \ldots, m_n\}$) within a reasonable range, the errors associated with rainfall forcing can be reduced. Compared with the additive errors model, the multiplicative errors model has an advantage that it does not depend on the scale of rainfall depths while it cannot correct

observed rainfall depths of zero (Kavetski et al., 2006; Vrugt et al., 2008).

Generally, the multipliers are assumed to subject to a Gaussian distribution with the variance ($\sigma_m$) and a constant mean ($\mu_m$). Based on the assumption that rain gauges tend to capture unbiased rainfall depths, the $\mu_m$ was set to 1. As to how to get the value of $\sigma_m$, please refer to Kavetski et at. (Kavetski et al., 2006) for details. In this study, we gave $\sigma_m$ a constant value according to the results after several iterations of the optimal procedure.

Substitute the input error model in Eq. 5 with the storm depth multiplier model, the Eq. 5 changes to:

$$p\big(\theta, m|\hat{Y}, \hat{\phi}\big) = c \cdot p(\theta) \cdot N(m|\mu_m, \sigma_m^2) \cdot \prod_{i=1}^{n} \frac{1}{\sqrt{2\pi}\sigma_e} exp\left(-\frac{\left(y_i(\theta, m|\hat{\phi})-\hat{y}_i\right)^2}{2\sigma_e^2}\right) \quad (6)$$

**2.4 DREAM**

DREAM, developed by (Vrugt et al., 2009a), is an adaptive MCMC algorithm to efficiently estimate the posterior probability density function of parameters in high-dimensional, complex sampling problems (Vrugt et al., 2009b). The





method explores global optimal samples by running multiple chains simultaneously and tuning the scale and orientation of the proposal distribution to the posterior distribution automatically (Vrugt et al., 2009b). The method shows excellent efficiency on multimodal target, highly nonlinear, complex distributions while maintaining ergodicity and detailed balance (Vrugt et al., 2009b). For the detailed theory and application procedures of the DREAM method, please refer to Vrugt et al. (2009a), Vrugt et al. (2008 ) and Vrugt et al. (2009b).

## 165 2.5 The linkage between DREAM and SWMM

DREAM and the SWMM model are connected within a Python platform. As described in Section 2.1, the SWMM model is packaged through Python language and some APIs used to obtain and change the parameter values of the SWMM model are developed and added to the model. With the APIs, the DREAM method can get and adjust the model parameters directly instead of rewriting the input file of the SWMM model frequently, which can save a lot of computing time. Besides, the 170 storm depth multiplier ( ) is integrated into the SWMM model to consider the rainfall uncertainty. The calibration procedure is as follows:

Step 1: Sample parameter values through the DREAM module.

Step 2: Replace the parameter values of the modified PySWMM with the sampled parameter values through the APIs.

Step 3: Run the modified PySWMM and return the output streamflow to the DREAM module.

Step 4: Expand the Markov chain through the DREAM algorithm and check if the Markov chain meets the termination condition. If the termination condition is met, stop the calibration, otherwise, return to Step 1.

The structure and workflow of the calibration framework are shown in Figure 1.

## 3 Case study

### 3.1 Study area and data collection

A small commercial area, located in Zhihuicheng, Guangzhou, China, is adopted for the application study. Zhihuicheng is located in the northeast of Tianhe district of Guangzhou. The climate of this area is dominated by subtropical monsoons, with an average annual precipitation of 1650 mm and an annual mean temperature of 21.8 ℃. More than 80% of the annual precipitation occurs in the period from April to September, leading to more flooding in this area. The study area has a relatively flat terrain, with a total area of 11.37 hm$^2$. The main land use type of this area is commercial land, accounting for 185 about 70%. The detail of the study area is shown in Figure 2.

A self-recorded tipping rain gauge was placed on the roof of the tallest building located in the southwest of the study area to collect rainfall data. The accuracy of the tipping rain gauge is 0.2 mm. An ultrasonic flowmeter with the accuracy of 2%FS (full scale) was placed in the manhole near the outfall of the drainage system of this area to collect runoff data. The





collection interval of the flowmeter is 1 min. Both the two devices are manufactured by THWater
(http://www.thuenv.com/h-col-103.html). The specific locations of the rain gauge and the flowmeter are shown in Figure 2.
Eight rainfall events and the corresponding streamflow are selected for the case study. The details of the rainfall events are
shown in Table 1. The drainage system (Figure 3) is obtained from the local government and is checked by a field
investigation. The land use data and the soil type are obtained through the field investigation. The main land use of this area
is commercial land while the main soil type is clay.

## 3.2 Model building

The drainage system of the study area is represented by 29 junctions (each junction for a manhole), 1 outfall, and 30 conduits.
According to the topography and the distribution of manholes, the study area is manually divided into 34 sub-catchments.
SWMM provides many different infiltration methods and routing methods. In this study, the Green-Ampt method is used to
model the infiltration process in the study area while the dynamic wave method (a complete solution of the one-dimensional
Saint-Venant flow equation) is used to model the routing process. The spatial distribution of the drainage system and the
sub-catchments is shown in Figure 3.

### 3.3 parameter sensitive analysis

Parameter sensitive analysis is usually performed before calibration to reduce the number of the parameters considered
during calibration. Sensitivity analysis is used to identify the impact of different model parameters on the corresponding
outputs and thus to distinguish influential parameters from non-influential parameters (Behrouz et al., 2020; Niazi et al.,
2017). In this study, the Morris method is used for the parameter sensitivity analysis. The Morris method identifies global
sensitivity from local derivatives which are sampled on a specific grid throughout the space of the parameters. The method is
based on one-at-a-time (OAT) methods. Each parameter is perturbed along a grid of a specific size to create a trajectory
through the parameter space (Herman et al., 2013; Morris, 1991).
According to the sensitivity analysis, 9 parameters are regarded as sensitive ones in the study area. It should be noted that the
parameter "width" is represented by Eq. 7 to reduce the dimension of the parameter set while remaining the spatial variation
of the "width" parameter across different sub-catchments (Inc., 2005). The detail of these parameters is shown in Table 2.

$$W = K \cdot \sqrt{A} \tag{7}$$

where $W$ represents the width of the sub-catchment; $A$ represents the area of the corresponding sub-catchment; $K$ is a scale
factor which needs to be calibrated.

Note: Initial deficit is a model state rather than a model parameter. However, since it has an impact on event-based
simulations, we take it as a calibration variable in this study.

3.4. Calibration strategy

The first 6 rainfall events and their corresponding streamflow of the outfall are used to calibrate the model, and the last 2
rainfall events and their corresponding streamflow are used to verify the calibrated model. Two calibration approaches are





used to evaluate the impact of input uncertainty on model calibration: the approach only considering parameter uncertainty and the approach considering both parameter uncertainty and input uncertainty. Specifically, the first one only calibrates the inherent parameters of SWMM while the latter calibrates the inherent parameters of SWMM and the storm depth multipliers. According to the land use, the sub-catchments are divided into two categories including commercial land sub-catchments

and green land sub-catchments. The value of the parameter "%Imperv" is given according to the sub-catchment category (i.e. the same value for the same sub-catchment category), considering the large difference of this parameter between the two sub-catchment categories. Other parameters are considered to be the same across different sub-catchments since the study area is relatively small and simple and these parameters vary little in the study area. After calibration, a total number of $2d$ (twice the number of calibrated parameters) Markov chains are obtained and the last one-third of the samples in each chain

are used to summarize the marginal densities of parameters and generate simulated outputs.

The objective of the calibration is to maximize the posterior probability of $\theta$, described in Eq. 6, and thus to minimize the sum of the squared residuals between simulated streamflow and observed streamflow. The Nash-Sutcliffe index ($E_{NS}$) is used to evaluate the efficiency of simulated streamflow, which is shown in Eq 8.

$$E_{NS} = 1 - \frac{\sum_{t=1}^{n}(q_{t,obs} - q_{t,sim})^2}{\sum_{t=1}^{n}(q_{t,obs} - \overline{q_{obs}})^2} \tag{8}$$

where $q_{t,obs}$ and $q_{t,sim}$ represent the observed streamflow and simulated streamflow, respectively; $\overline{q_{obs}}$ represents the average of the observed streamflow; $t$ represents the time step of the streamflow sequence and $n$ represents the total number of the runoff sequence.

## 4 Results and discussion

### 4.1 Calibration only considering parameter uncertainty

The estimation of model parameters was firstly conducted only considering the parameter uncertainty. The posterior marginal probability density distributions of the calculated parameters are presented in Figure 4. The results show that most estimations of parameters K, %Imperv for commercial land, %Imperv for green land, and Dstore-Imperv are located in a relatively narrow interval included in the individual prior range of the parameters, which indicates that these parameters are more sensitive than others in the study area and the DREAM method is capable to find reasonable parameter values. It

should be noted that most parameters are approximately Gaussian except parameters %Imperv for commercial land, %Imperv for green land and N-imperv. The posterior marginal probability distributions of these parameters are significantly departed from the normal distribution, and most probability mass is concentrated at the upper boundaries of these parameters. It implies that the more fitted values for these parameters may beyond the physically realistic range. The contradiction between the optimal parameter values and the physical limitations is probably ascribed to the

representativeness of the parameter itself, the structural deficiencies of SWMM, and the errors in the input data. The simulated and observed runoff of the outfall are compared in Figure 5. The Nash-Sutcliffe indices of the simulations can be





found in Figure 6. In general, the calibrated SWMM model can capture the fluctuation characteristics of the runoff well for most rainfall events. The medians of the Nash-Sutcliffe indices are larger than 0.55 for most rainfall events except rainfall event 20180831. Figure 5 shows that the relatively low Nash-Sutcliffe index is caused by the obvious underestimation of the

streamflow after the peak flow. It may be caused by the measurement errors of runoff or domestic sewage discharged into the drainage system since there is little rainfall during this period. Moreover, the hydrographs of the simulated runoff in Figure 5 show that the simulated runoff is not distributed as much as that the posterior marginal probability distributions of model parameters imply. This confirms the existence of the phenomenon of equifinality in the SWMM model. It can also be found from the hydrographs that the calibrated SWMM model underestimates the peak flow for most rainfall events. This

may lead to the failure of flood control facilities designed with the SWMM model calibrated only considering the parameter uncertainty.

**4.2 Calibration considering both parameter uncertainty and input uncertainty**

The posterior marginal probability density distributions of the selected model parameters and storm depth multipliers are shown in Figure 7. Unlike the posterior marginal probability density distributions of parameters calibrated only considering

parameter uncertainty, the posterior marginal probability density distributions of all the parameters calibrated considering both parameter uncertainty and input uncertainty are approximately Gaussian. It means that the optimal values of all the parameters are located in physically reasonable ranges. The differences between the posterior marginal probability density distributions of parameters calibrated through the two different calibration approaches imply that the unrealistic values of the parameters %Imperv for commercial land, %Imperv for green land and N-imperv obtained from the first calibration

approach are caused by compensating for errors in the rainfall data. Further comparison between parameter values obtained from the two different calibration approaches is shown in Figure 8. The results show that most parameter values show a significant difference between the two calibration approaches except the parameter N-Perv. The exception of the parameter N-Perv may be attributed to that the simulated runoff is not very sensitive to this parameter in the study area since the proportion of pervious areas in the study area is relatively small. The posterior marginal probability density distributions of

storm depth multipliers are shown in Figure 9 and Figure 10. Most of the storm depth multipliers are approximately Gaussian distribution except the storm depth multipliers for rainfall events during the validation period, which indicates that the DREAM method can define storm depth multipliers well. The storm depth multipliers of rainfall events during the validation period are randomly sampled from the storm depth multiplier samples of rainfall events during the calibration period, which leads to the abnormal distributions of the storm depth multipliers during the validation period. It can be found

from Figure 10 that medians of storm depth multiplier samples range from 1.2 to 2.0, which means that the rain gauge tends to underestimate the actual rainfall depth obviously in the study area. This is not consistent with the result obtained by Vrugt (Vrugt et al., 2008) in a study about input uncertainty in hydrological modeling of natural watersheds that most of the storm depth multipliers are distributed around 1. Some studies indicated that the precipitation data measured by tipping bucket rain gauges can be largely affected by wind field nearby (Dotto et al., 2014). To explore the relationship between storm depth





multipliers and the wind field nearby, the corresponding wind speeds of the rainfall events are shown in Table 3. Most rainfall events show that higher wind speed leads to greater storm depth multiplier, which indicates that the effect of the wind field is an important source of rainfall errors. It should be noted that the underestimation of rainfall depth in this study is more serious than that in other studies. The reason may be that the rainfall gauge is placed on the roof of a building where turbulence may be intensified by the wind blocking effect of the building, which reduces the catch efficiency of rainfall

gauges significantly. The antecedent soil moisture is another factor that can affect the storm depth multipliers since it determines how much rainfall can be absorbed by the soil. However, the studied area is dominated by impervious areas, the impact of the antecedent soil moisture is considered to be small.

The runoff simulated with considering both parameter uncertainty and input uncertainty is shown in Figure 11, with the observed runoff as the reference. The results show that the runoff simulated with considering both parameter uncertainty and

input uncertainty captures peak flows much better than that only considering parameter uncertainty, especially for the validation period. This indicates that the calibration approach considering both parameter uncertainty and input uncertainty is more suitable for runoff prediction which is the basis of the designing of urban flooding control facilities. Figure 6 shows the comparison of Nash-Sutcliffe indices of runoff simulations obtained through the two different calibration approaches. Most of the Nash-Sutcliffe indices of simulations obtained from the model calibrated considering both parameter uncertainty and

input uncertainty are greater than that obtained from the model calibrated only considering parameter uncertainty. Although the Nash-Sutcliffe indices of runoff simulations considering both parameter and input uncertainty are less than that only considering parameter uncertainties for events 20180530 and 20180724, they are also acceptable. Moreover, the optimal model parameter values calibrated considering both parameter uncertainty and input uncertainty are more physically realistic than that calibrated only considering parameter uncertainty. It can be concluded that calibration considering both parameter

uncertainty and input uncertainty is more robust than that only considering parameter uncertainty.

## 5 Conclusions

An automatic calibration framework of SWMM based on Bayesian theory, simultaneously considering uncertainties of parameter and rainfall, was developed. The framework couples DREAM and modified SWMM through the newly developed API functions built in SWMM to obtain and adjust parameter values of SWMM models. A rainfall error model, featured by

storm depth multiplier, was also integrated into the framework to consider the systematic rainfall errors. The framework can calibrate SWMM models according to the specific criterion defined by the user. A case study in Guangzhou, China was conducted to demonstrate the use of the calibration framework. The calibration capability of the framework was tested and the impacts of rainfall uncertainty on model parameter estimations and simulated runoff boundaries were identified. The main conclusions are as follows: (1) the newly developed framework can obtain relatively reasonable parameter values of

SWMM models; (2) ignoring rainfall uncertainty may lead to unrealistic estimations of model parameters; (3) parameter values estimated when considering both parameter uncertainty and rainfall uncertainty are well defined within their

physically realistic ranges in the study area; (4) rainfall intensity is obviously underestimated in the study area according to the storm depth multipliers estimated by the framework; (5) calibration considering both parameter uncertainty and rainfall uncertainty captures peak flow much better and is more robust in terms of the Nash Sutcliffe index than that only considering

parameter uncertainty.

Although the proposed auto-calibration framework has been demonstrated effective, it has some weakness that can be improved further. For example, the storm depth multiplier model used to quantify the rainfall uncertainty is relatively simple. It does not consider the measurement errors of rainfall patterns. More complicated rainfall error model can be incorporated into the proposed framework to quantify the impact of rainfall uncertainty on the simulations more accurately. Besides, the

structure uncertainty of the SWMM model is not considered in this study since many studies have confirmed the stability of SWMM. However, the parameters and modeling results can be affected by the method to set up the model, such as the division of sub-basins, the choice of the infiltration model, and the choice of the pipe routing method. In further studies, the impact of the structure uncertainty caused by the method to set up the model should be investigated.

*Data availability.* All the data are available on GitHub (https://github.com/pandagxc/autocalibration-of-SWMM).

*Author contributions.* X.G.: conceptualization, methodology, software, validation, formal analysis, investigation, writing—original draft preparation, visualization. Z.Y.: conceptualization, resources, writing—review and editing, supervision, project administration, funding acquisition. D.H.: methodology, validation, data curation, writing—review and editing. G.H.: software, writing—review and editing Q.Z.: data curation, visualization.


*Competing interests.* The authors declare that they have no conflict of interest.

*Acknowledgements.* Thanks to the authors of Matplotlib for providing an instrument for the visualization works. The authors would like to extend great thanks to the anonymous reviewers, who provide many valuable pieces of advice for the improvement of the paper.


*Financial support.* This study is financially supported by the National Key Research and Development Project (No. 2016YFC0402707), the National Natural Science Foundation of China (51879274 and 51739011).

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

**Table 1. Detail of the rainfall events**

| Events | Date | Duration (min) | Rainfall depth (mm) |
|---|---|---|---|
| 20180530 | 2018/5/30 | 44 | 29 |
| 20180622 | 2018/6/22 | 50 | 27.8 |
| 20180625 | 2018/6/25 | 234 | 49.8 |
| 20180713 | 2018/7/13 | 17 | 8.4 |
| 20180724 | 2018/7/24 | 49 | 16.4 |
| 20180828 | 2018/8/28 | 350 | 61.4 |
| 20180831 | 2018/8/31 | 301 | 39.6 |
| 20190825 | 2019/8/25 | 255 | 58.6 |

**Table 2. Sensitive parameters**



| Parameters | Description (unit) | Range | Data source |
|---|---|---|---|
| N-Imperv | Manning's roughness of impervious area | [0, 0.03] | Zeng jiajun et. Al (Zeng et al., 2020) |
| N-Perv | Manning's roughness of pervious area | [0.03, 0.80] | Zeng jiajun et. Al (Zeng et al., 2020) |
| Dstore-Imperv | Depth of depression storage on impervious area (mm) | [1, 3] | Zeng jiajun et. Al (Zeng et al., 2020) |
| Dstore-Perv | Depth of depression storage on pervious area (mm) | [2, 8] | Zeng jiajun et. Al (Zeng et al., 2020) |
| %Slope | Average surface slope | [0, 0.03] | Field investigation |
| K (Width) | Width factor of overland flow path | [0, 5] | InfoSWMM user manual (Inc., 2005) |
| %Imperv | Percent of impervious area | [0.50, 0.80] for commercial land [0, 0.20] for green land | SWMM user's manual (Rossman and Huber, 2016) |
| Conductivity | Soil saturated hydraulic conductivity (mm/hr) | [1, 200] | Zeng jiajun et. Al (Zeng et al., 2020) |
| Initial deficit | Difference between soil porosity and initial moisture content | [0, 0.50] | SWMM user's manual (Rossman and Huber, 2016) |

**Table 3. Wind speeds corresponding to the studied rainfall events.**

| Events | Storm depth Multiplier | Wind speed (m/s) |
|---|---|---|
| 20180530 | 1.26 | 2.7 |
| 20180622 | 1.56 | 3.0 |
| 20180625 | 1.55 | 3.0 |
| 20180713 | 1.90 | 4.5 |
| 20180724 | 1.21 | - |
| 20180828 | 1.84 | 2.5 |

Note: "-" means the data is missing.



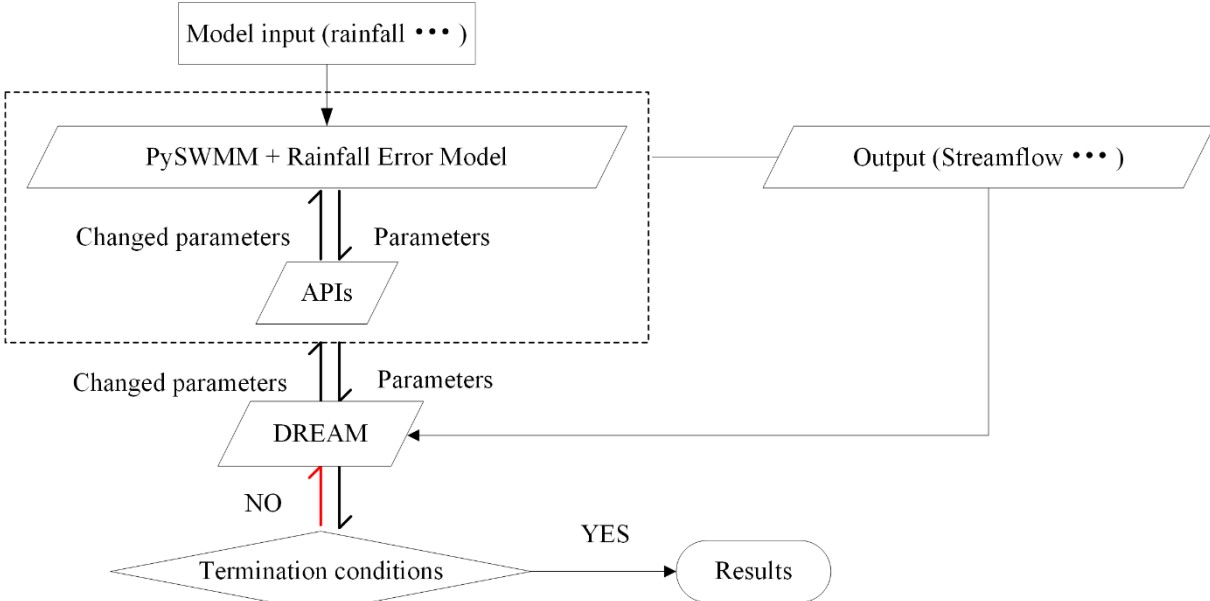

**Figure 1. The structure and workflow of the developed calibration framework.**

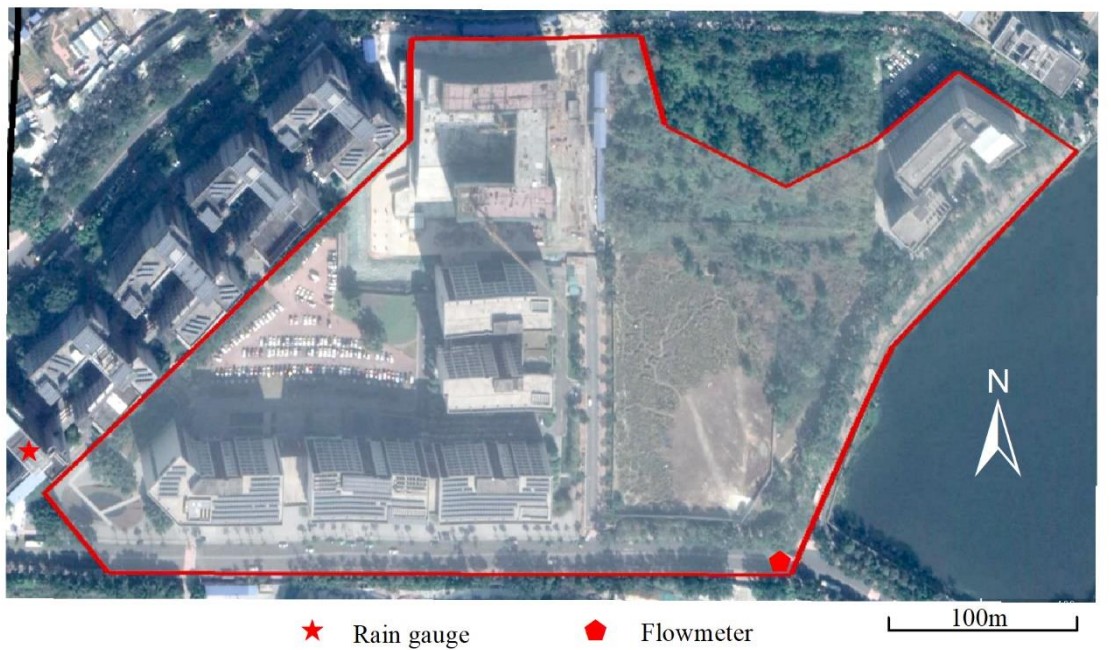

**Figure 2. The map of the study area and locations of monitoring equipment. The area within the red line is the study area. Map data: © Google, Maxar Technologies.**





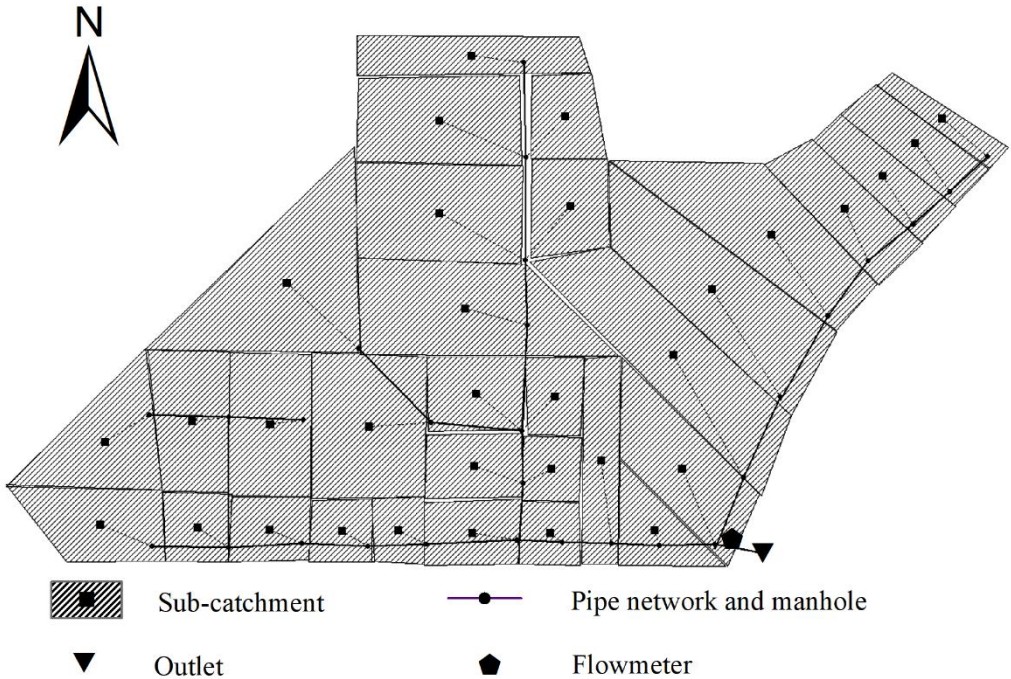

**Figure 3. Watershed overview in the SWMM model.**

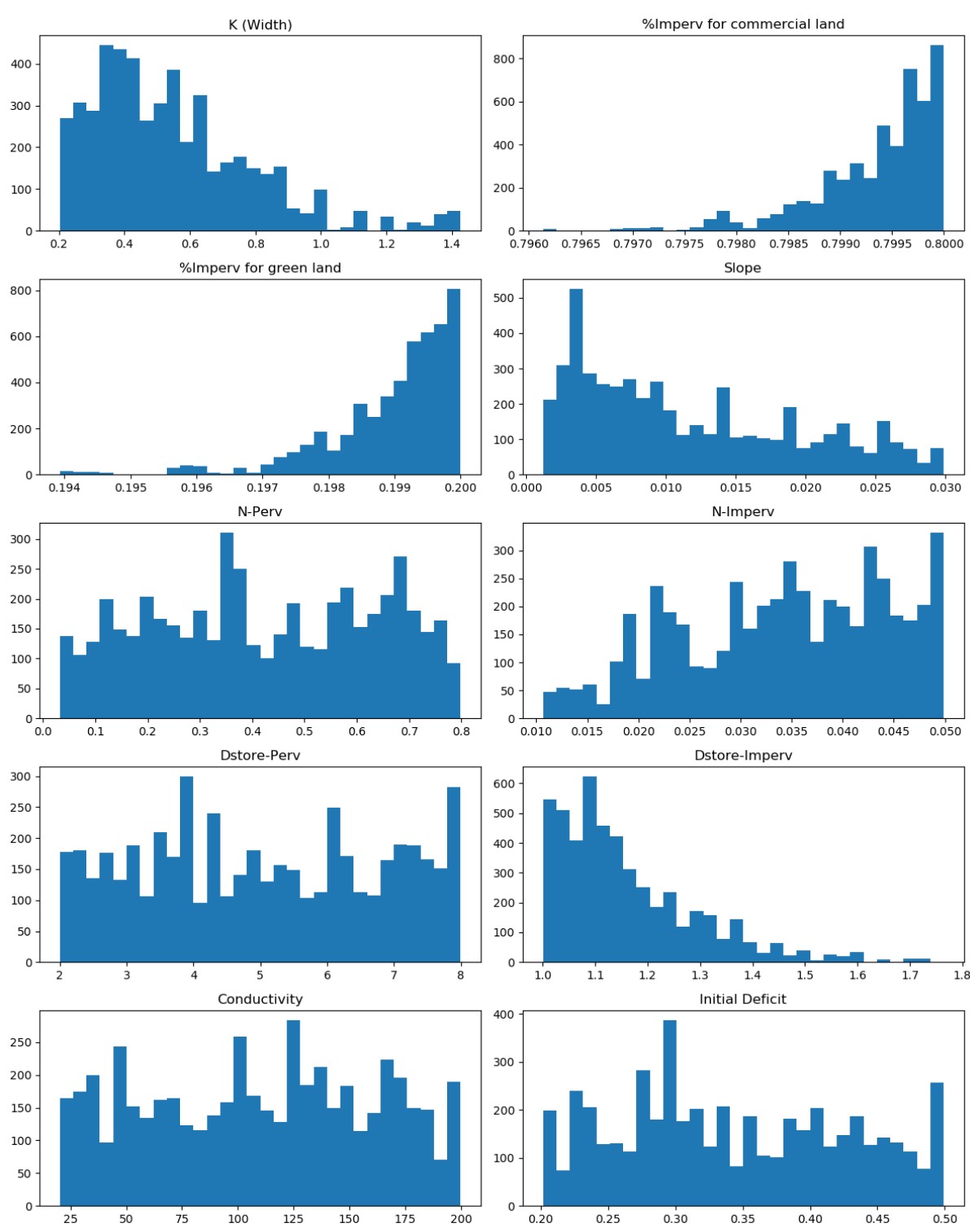

**Figure 4. The posterior marginal probability density distributions of parameters only considering parameter uncertainty.**







**Figure 5. Comparison between the observed runoff and simulated runoff using the model calibrated only considering parameter**
**uncertainty. The shadow of the black line represents the 95% uncertainty range caused by parameter uncertainty.**





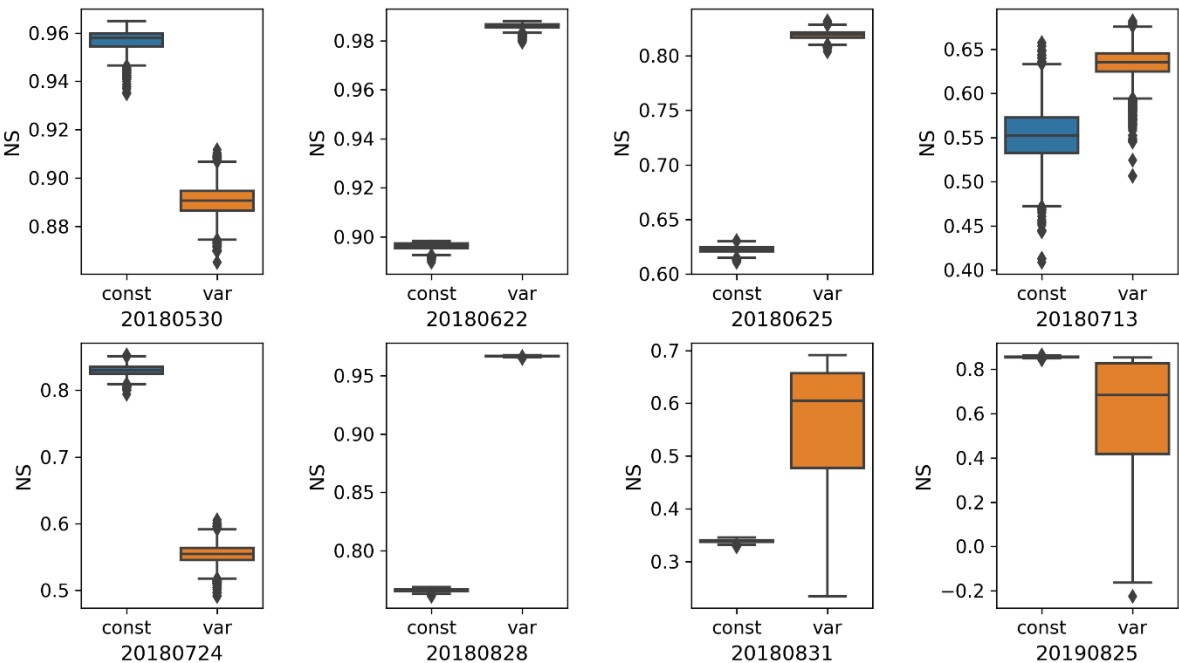

**Figure 6.** Comparison of Nash-Sutcliffe indices of the simulations obtained through the two different calibration approaches. The boxes indicate the 25th, 50th, and 75th percentiles of the scaling rates, and the vertical lines indicate the 5th and 95th percentiles. Black markers refer to values greater than 1.5 times the interquartile range away from the bottom or top of the box.










**Figure 7. The posterior marginal probability density distributions of parameters calibrated considering both parameter uncertainty and input uncertainty.**

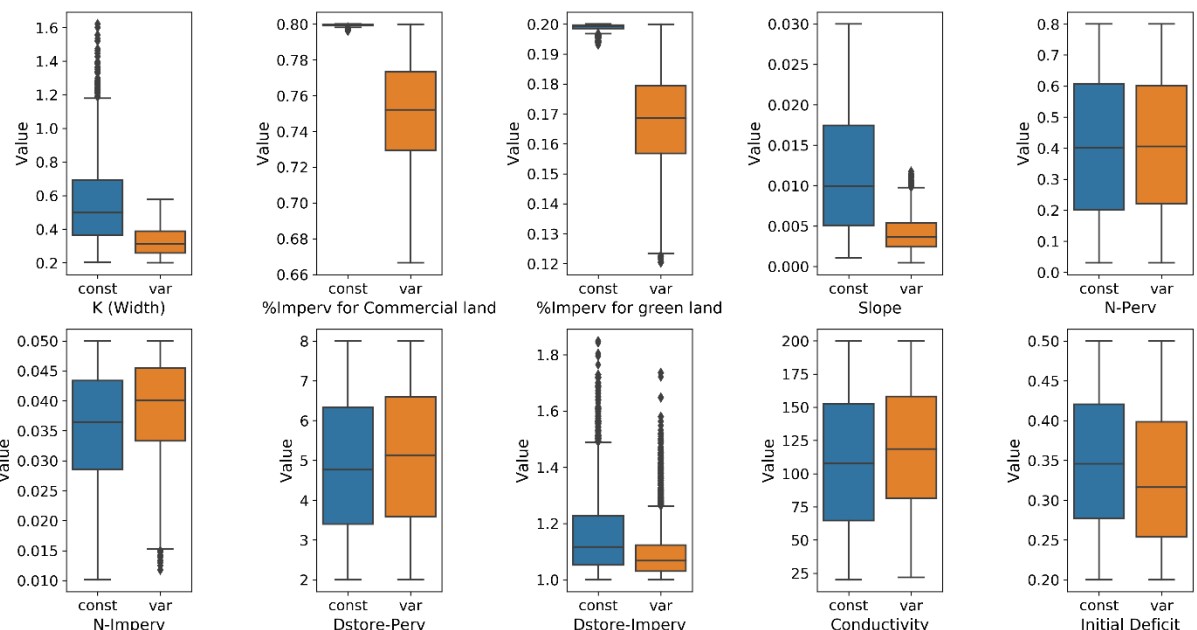

**Figure 8. Comparison of parameter values obtained from the two different calibration approaches. The boxes indicate the 25th, 50th, and 75th percentiles of the scaling rates, and the vertical lines indicate the 5th and 95th percentiles. Black markers refer to values greater than 1.5 times the interquartile range away from the bottom or top of the box.**



**Figure 9. The posterior marginal probability density distributions of storm depth multipliers.**

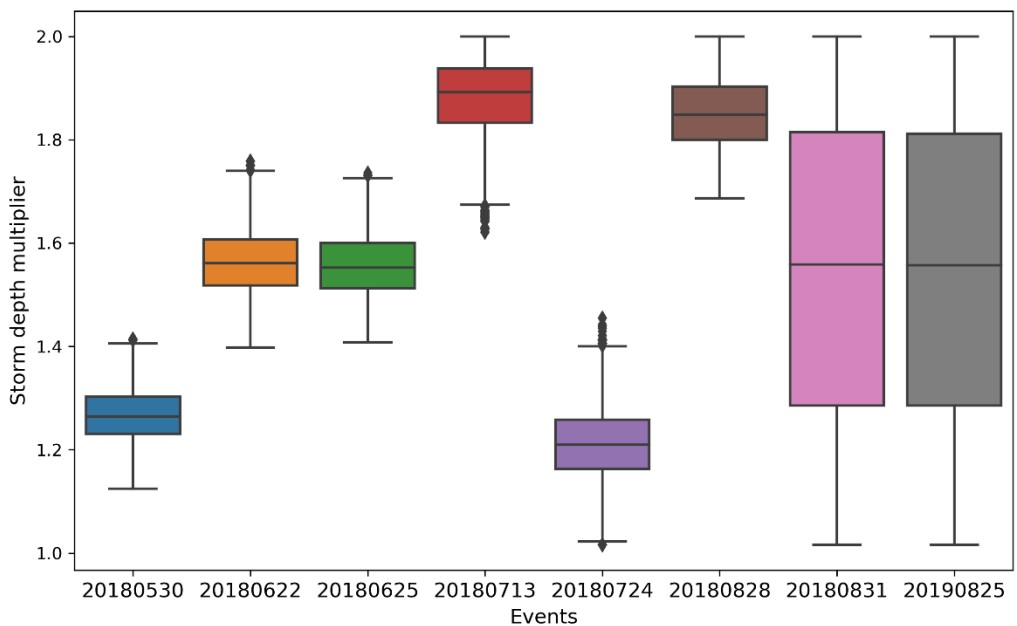

**Figure 10. Comparison of storm depth multipliers for different rainfall events. The boxes indicate the 25th, 50th, and 75th percentiles of the scaling rates, and the vertical lines indicate the 5th and 95th percentiles. Black markers refer to values greater than 1.5 times the interquartile range away from the bottom or top of the box.**





**Figure 11. Comparison between the observed runoff and simulated runoff using the model calibrated considering both parameter uncertainty and input uncertainty. The shadow of the black line represents the 95% uncertainty range caused by parameter uncertainty and input uncertainty.**