# Peer review of "A Framework for Automatic Calibration of SWMM Considering Input Uncertainty"

_Hydrology and Earth System Sciences, 2020_

## Referee Comment (RC1) · Edward Tiernan (Referee) · 4 Nov 2020

General Comments:

This article correctly identifies a knowledge gap in rainfall-runoff models wherein calibration typically considers uncertainty only in the model's parameters, neglecting the uncertainty in forcing data. Additionally, to the best of my knowledge, tying the DREAM calibration methodology to SWMM using a modified PySWMM API is a new and useful contribution. For these reasons, the research that the author's propose is of great interest to the hydrologic community.

However, the work done in this paper towards addressing that knowledge gap is unconvincing. The authors aim to "settle [the] problem" of rainfall uncertainty being left

out of the calibration process. Yet, at the end of the paper the problem remains un-settled and their arguments undefended. Deficiencies in the simplistic case study and calibration approach call into question the conclusions that the author's method defini-tively demonstrates the dominance of forcing data uncertainty in the calibration pro-cess. The rainfall uncertainty model itself does not provide convincing evidence for the true magnitude of the rainfall uncertainty issue; their calibrated rainfall model implies an underestimation of rain depth greater even than the worst case described in the lit-erature. Given the implications of the assertions posed in the introduction, and the lack of convincing evidence to support those assertions, the contribution of this research to the hydrologic community is hard to justify.

Specific Comments:

In Eq. 3 the authors offer the Sum of Squared Residuals as a common objective function for minimizing the residual. In the results, though, the authors swapped the SSR in favor of the Nash-Sutcliffe Efficiency Index. Why? The Nash-Sutcliffe Index is well known to have deficiencies for optimizing hydrograph behavior, and the author's calibration attempts in the case study seem to fail both this metric and the eye test. I would recommend using more than one objective function to help ameliorate the drawbacks of calibrating to any one function. The authors admit that the peak flow was poorly captured by the "parameter-only" calibrated model; it may be beneficial for another objective function to be targeted at that hydrograph feature.

Peak Difference=|Peak_sim - Peak_obs|

The poor calibration of a relatively simple case study system does much to undermine the researcher's conclusions on the impact of the rainfall uncertainty. Another option would be to allow for parameters to vary for each subcatchment independently. This increases the dimensionality of the calibration problem 34-fold, but the MCMC method-ology within DREAM, and the pre-processing sensitivity analysis, make this reasonably achievable.

The preponderance of my experience in calibrating SWMM has been focused on the subcatchment parameter set (credit to the authors for identifying the myopia of the community on that front). However, I find it hard to believe that the two options for considering rainfall uncertainty are an additive correction factor and a multiplicative one. This simplistic approach feels like less of a foray into an unexplored frontier of calibrating rainfall-runoff models, and more of just adding another parameter to be calibrated in the same manner as any other.

From my understanding, a narrow posterior distribution for a parameter is evidence of a high confidence in that parameter's optimal value. The conclusion from Fig 4 that "most parameters are approximately Gaussian" is puzzling. Rather than relying on the "approximately" disclaimer, why wasn't a statistical test to demonstrate Gaussian-ness performed.

I remain vexed as to the purpose of Figures 6 and 8. I wish they were explained in the text rather than just referenced. I don't have any clue what constant vs varied mean in either Figure's context. The authors claim that the narrowness of the 95% confidence interval band around the median hydrograph "confirms the existence of equifinality". That seems logically inconsistent. Equifinality describes the phenomenon that multiple parameter sets might yield the same hydrograph. It does not suggest that every parameter set will yield the same hydrograph. If anything, that observation calls into question the initial sensitivity analysis.

The general aim of this paper was to demonstrate that the uncertainty introduced by forcing data is a significant contributor to the uncertainty in system behavior. However, seeking to prove that by showing the improvement of a calibrated model by adding a rainfall multiplier as another calibration parameter presents a catch-22. The better that the initial calibration is, the less impactful the rainfall calibration will seem. The author's conclusions are better supported the less carefully the system's parameters are calibrated. It would be a much more compelling paper structure to compare the calibration refinement from a rainfall uncertainty model vs another weakly studied uncertainty

source.

Technical Corrections:

To restate a previous question as a suggestion. Consider using metrics beyond the NSE to evaluate the calibrated solutions.

In Section 2.5, the linkage between DREAM and SWMM is described. The steps enumerated don't add any description value to Figure 1. Rather than a regurgitation of the Figure, I'd be interested to see some of the other questions I had be answered in the delineation of this workflow. Such as, is there any difference in the workflow for considering just the subcatchment parameters and the combined forcing/subcatchment parameter sets?

There are a number of capitalization/grammar mistakes. Such as The last sentence in the abstract. Capitalize "The" "3.3 parameter sensitive analysis" should be "3.3 Parameter Sensitivity Analysis"

The parenthetical citations seem... wrong.

---

## Referee Comment (RC2) · Anonymous Referee #2 · 7 Dec 2020

The study investigates the effects of rainfall errors and parameter uncertainty on SWMM rainfall-runoff simulation and calibration. It brings together existing methodologies (SWMM, DREAM, rainfall error model) and presents an application for a small urban catchment in China.

This is an interesting topic but I have several critical comments on the current manuscript.

-novelty: the presented methodology is not really new, but is mostly an application of existing methodologies to a SWMM modeling case study. The paper would be much more novel if the results would be used to gain new insights about the system's response and test additional model improvements. In other words, also bring in structural model errors, since they appear to be significant in the case study (see below).

-the main conclusion of the paper is that accounting for rainfall errors improves simulation of rainfall-runoff response with SWMM. For example on line 295: "The results show that the runoff simulated with considering both parameter uncertainty and input uncertainty captures peak flows much better than that only considering parameter uncertainty, especially for the validation period." This conclusion is not supported by the results in fig. 5 and 11, which show mixed results, i.e. both better and worse performance after accounting for rainfall errors.

-several modeling assumptions are not met. For example, the residuals are assumed iid gaussian. This assumption is clearly violated in figs. 5 and 11, which show significant systematic deviations between simulated and observed discharge, that indicate one or more aspects of the system are not captured in the model. Another example relates to assumed unbiasedness of the rainfall measurements, (line 152). Results of the case study show that estimated rainfall multipliers are all greater than 1 (fig. 9). So at least for the case study, the unbiasedness assumption needs to be revised. Indeed, if, as suggested by the authors, rainfall errors are caused by wind-related undercatch, then one can expect systematic underestimation of rainfall.

-line 154: it sounds like the optimal value of sigma_m was obtained "manually", but such manual calibration was criticized earlier in the introduction. Why not automate the estimation of sigma_m? Either by jointly estimating sigma_m together with the other parameters, or if possible, integrating out sigma_m from the posterior before running MCMC.

-how was sigma_e (residual standard error) in eq.6 estimated?

-line 170: empty parentheses, missing equation?

-the case study considers a very small catchment, is spatial variability of rainfall an important source of rainfall error here (as suggested in the introduction)?

-line 191: why are these rainfall events selected and why so few? The small number of

events considered here limits robustness of any conclusions drawn from the analysis.

---

## Author Comment (AC1) · 4 Mar 2021

General Comments: This article correctly identifies a knowledge gap in rainfall-runoff models wherein calibration typically considers uncertainty only in the model's parameters, neglecting the uncertainty in forcing data. Additionally, to the best of my knowledge, tying the DREAM calibration methodology to SWMM using a modified PySWMM API is a new and useful contribution. For these reasons, the research that the author's propose is of great interest to the hydrologic community. However, the work done in this paper towards addressing that knowledge gap is un- convincing. The authors aim to "settle [the] problem" of rainfall uncertainty being left out of the calibration process. Yet, at the end of the paper the problem remains unsettled and their arguments un-defended. Deficiencies in the simplistic case study and calibration approach call into

question the conclusions that the author's method definitively demonstrates the dominance of forcing data uncertainty in the calibration process. The rainfall uncertainty model itself does not provide convincing evidence for the true magnitude of the rainfall uncertainty issue their calibrated rainfall model implies an underestimation of rain depth greater even than the worst case described in the literature. Given the implications of the assertions posed in the introduction, and the lack of convincing evidence to support those assertions, the contribution of this research to the hydrologic community is hard to justify. Specific Comments:

Reply: Yes, you are right. the reason for the dominance of forcing data uncertainty is that the model structural uncertainty was not considered. In the revised manuscript, we added a first-order autoregressive model to represent the autocorrelation of the residuals and thus to consider the structural uncertainty of the SWMM model. The uncertainty associated with forcing data decreased obviously after considering the model structural uncertainty. See Figure 1, thank you.

In Eq. 3 the authors offer the Sum of Squared Residuals as a common objective function for minimizing the residual. In the results, though, the authors swapped the SSR in favor of the Nash-Sutcliffe Efficiency Index. Why? The Nash-Sutcliffe Index is well known to have deficiencies for optimizing hydrograph behavior, and the author's calibration attempts in the case study seem to fail both this metric and the eye test. I would recommend using more than one objective function to help ameliorate the drawbacks of calibrating to any one function. The authors admit that the peak flow was poorly captured by the "parameter-only" calibrated model; it may be beneficial for another objective function to be targeted at that hydrograph feature. Peak Difference=|Peak_sim - Peak_obs|

Reply: Yes, you are right. Peak flow differences, total streamflow differences as well as squared residuals were used as objective functions for the calibration and the Nash-Sutcliffe Efficiency index, the peak flow bias and the total streamflow bias were used to evaluate the calibration efficiency in the revised manuscript. Thank you.

The poor calibration of a relatively simple case study system does much to undermine the researcher's conclusions on the impact of the rainfall uncertainty. Another option would be to allow for parameters to vary for each subcatchment independently. This increases the dimensionality of the calibration problem 34-fold, but the MCMC methodology within DREAM, and the pre-processing sensitivity analysis, make this reasonably achievable.

Reply: Yes, you are right. However, those parameters with clear physical meanings, such as the surface slope and Manning's roughness, can be regarded as the same between different subcatchments with the same land use type since the study area is relatively small. Therefore, we still kept these parameters the same for subcatchments with the same land use type considering the efficiency of computation. The parameter "width" is sensitive to the shape of the subcatchment, so we allowed this parameter to vary for subcatchments with different shapes in the revised manuscript. Specifically, the subcatchments of the study area in this manuscript were divided into 8 types according to their shapes, so there are 8 parameters for the parameter "width" in the revised manuscript. Thank you.

The preponderance of my experience in calibrating SWMM has been focused on the subcatchment parameter set (credit to the authors for identifying the myopia of the community on that front). However, I find it hard to believe that the two options for considering rainfall uncertainty are an additive correction factor and a multiplicative one. This simplistic approach feels like less of a foray into an unexplored frontier of calibrating rainfall-runoff models, and more of just adding another parameter to be calibrated in the same manner as any other.

Reply: Yes, you are right. The storm depth multiplier model is relatively simple. But as far as I know, most of the studies about input uncertainties use this multiplicative model to identify the rainfall uncertainty and get reasonable results. I guess the main reason why you think this model is not credible is the large underestimation of the rainfall depth the model shows. However, one reason for the large errors of rainfall depths may be

that we did not consider the structure uncertainty of the SWMM model and some errors caused by model deficiencies were integrated into the rainfall depth errors; the other reason may be that rainfall intensity is more easily underestimated in urban areas since the measurement of rainfall intensity by rain barrels can be influenced by turbulent flow near the orifice caused by urban wind and shading of rainfall by buildings. In the revised manuscript, the model structure uncertainty was considered. The results illustrated that uncertainties associated with rainfall decrease after considering the model structural uncertainty. See Figure 1, thank you.

From my understanding, a narrow posterior distribution for a parameter is evidence of a high confidence in that parameter's optimal value. The conclusion from Fig 4 that "most parameters are approximately Gaussian" is puzzling. Rather than relying on the "approximately" disclaimer, why wasn't a statistical test to demonstrate Gaussian-ness performed.

Reply: Yes, you are right. a narrow posterior distribution for a parameter means a high confidence in the parameter's optimal value, which implies that the calibration method can find reasonable values of parameters. Besides, all the parameters are calibrated in a same Bayesian inference framework, which means that they have the same chance to get their reasonable values, so the narrower the parameter distribution, the more sensitive the parameter. Thank you.

I remain vexed as to the purpose of Figures 6 and 8. I wish they were explained in the text rather than just referenced. I don't have any clue what constant vs varied mean in either Figure's context. The authors claim that the narrowness of the 95% confidence interval band around the median hydrograph "confirms the existence of equifinality". That seems logically inconsistent. Equifinality describes the phenomenon that multiple parameter sets might yield the same hydrograph. It does not suggest that every parameter set will yield the same hydrograph. If anything, that observation calls into question the initial sensitivity analysis.

[Figure]

Reply: Sorry, the evidence of the equifinality is not explained clearly in the manuscript. In the revised version, we added some explanations about the equifinality as follows: "The figure shows that the uncertainty of the simulated runoff is not as much as that the posterior marginal probability distributions of the model parameters imply. In other words, the uncertainty of the parameters is greater than that of the simulated runoff, which means that there are some different parameter sets having similar simulated results. This confirms the existence of the phenomenon of equifinality in the SWMM model." Thank you.

The general aim of this paper was to demonstrate that the uncertainty introduced by forcing data is a significant contributor to the uncertainty in system behavior. However, seeking to prove that by showing the improvement of a calibrated model by adding a rainfall multiplier as another calibration parameter presents a catch-22. The better that the initial calibration is, the less impactful the rainfall calibration will seem. The author's conclusions are better supported the less carefully the system's parameters are calibrated. It would be a much more compelling paper structure to compare the calibration refinement from a rainfall uncertainty model vs another weakly studied uncertainty source.

Reply: Yes, you are right. The identification of the influence of the rainfall uncertainty is controlled by the initial calibration. As you suggest, we added a structure uncertainty model into the calibration framework and compared the performance of the calibration framework with and without rainfall uncertainty in the revised manuscript. Thank you.

Technical Corrections: To restate a previous question as a suggestion. Consider using metrics beyond the NSE to evaluate the calibrated solutions.

Reply: The NSE, the peak flow bias, and the total streamflow bias were used to evaluate the calibrated solutions in the revised manuscript. Thank you.

In Section 2.5, the linkage between DREAM and SWMM is described. The steps enumerated don't add any description value to Figure 1. Rather than a regurgitation

of the Figure, I'd be interested to see some of the other questions I had be answered in the delineation of this workflow. Such as, is there any difference in the workflow for considering just the subcatchment parameters and the combined forcing/subcatchment parameter sets?

Reply: The workflow of different calibration frameworks (with and without the storm depth multiplier model) is similar. The multipliers of the storm depth multiplier model are sampled in the framework considering the rainfall uncertainty but set to be 1 in the framework not considering the rainfall uncertainty model. The workflow steps were rewritten as follows. Thank you.

"Step 1. The parameters of the integrated model, combining the SWMM model, AR-1 model, and storm depth multiplier model, are sampled through the DREAM module according to their prior probability distributions. If rainfall uncertainty is not considered, the values of all the storm depth multipliers will be set to be one. Step 2. The sampled parameter values are passed to the integrated model though the developed APIs. The streamflow is then simulated by the model and the posterior probabilities are calculated based on the simulated and observed streamflow. Step 3. The Markov chain is expanded through the DREAM algorithm according to the obtained posterior probabilities. The termination condition of the Markov chain is then checked. If the termination condition is met, stop the calibration, otherwise, return to Step 1."

There are a number of capitalization/grammar mistakes. Such as The last sentence in the abstract. Capitalize "The" "3.3 parameter sensitive analysis" should be "3.3 Parameter Sensitivity Analysis"

Reply: They have been revised, thank you.

The parenthetical citations seem. . . wrong.

Reply: It has been revised, Thank you.
* * *
367, 2020.

Storm depth multiplier 20180530

Storm depth multiplier 20180622

Storm depth multiplier 20180625

Storm depth multiplier 20180828

Storm depth multiplier 20180831

Storm depth multiplier 20190825

**Fig. 1.**

---

## Author Comment (AC2) · 4 Mar 2021

The study investigates the effects of rainfall errors and parameter uncertainty on SWMM rainfall-runoff simulation and calibration. It brings together existing methodologies (SWMM, DREAM, rainfall error model) and presents an application for a small urban catchment in China. This is an interesting topic but I have several critical comments on the current manuscript. -novelty: the presented methodology is not really new, but is mostly an application of existing methodologies to a SWMM modeling case study. The paper would be much more novel if the results would be used to gain new insights about the system's response and test additional model improvements. In other words, also bring in structural model errors, since they appear to be significant in the case study (see below).

Reply: we added a first-order autoregressive model to represent the autocorrelation of the residuals and thus to consider the structural uncertainty of the SWMM model in the revised manuscript. And deficiencies of SWMM are also discussed in the revised manuscript. Thank you.

-the main conclusion of the paper is that accounting for rainfall errors improves simulation of rainfall-runoff response with SWMM. For example on line 295: "The results show that the runoff simulated with considering both parameter uncertainty and input uncertainty captures peak flows much better than that only considering parameter uncertainty, especially for the validation period." This conclusion is not supported by the results in fig. 5 and 11, which show mixed results, i.e. both better and worse performance after accounting for rainfall errors.

Reply: Yes, you are right. The previous statement about the peak flow was not accurate. The figures about simulated flows show mixed results. To further evaluate the calibration results, we calculated the bias of peak flow (Figure 1) and total flow (Figure 2) in the revised manuscript. The performance of the two calibration approaches mixes in terms of peak flow bias, while the approach considering rainfall errors performs better than the approach not considering rainfall errors at least in the calibration period. Thank you.

-several modeling assumptions are not met. For example, the residuals are assumed iid gaussian. This assumption is clearly violated in figs. 5 and 11, which show significant systematic deviations between simulated and observed discharge, that indicate one or more aspects of the system are not captured in the model.

Reply: Yes, the assumption of the idd gaussian of the residuals is usually not realistic in hydrologic modeling. The time series of residuals are typically autocorrelated and nonstationary. We added a first-order autoregressive model to represent the autocorrelation of the residuals and thus to consider the structural uncertainty of the SWMM model in the revised manuscript. Besides, BOX-Cox transformation of the simulated

and observed streamflow is used to reduce heteroscedasticity. Thank you.

Another example relates to assumed unbiasedness of the rainfall measurements, (line 152). Results of the case study show that estimated rainfall multipliers are all greater than 1 (fig. 9). So at least for the case study, the unbiasedness assumption needs to be revised. Indeed, if, as suggested by the authors, rainfall errors are caused by wind-related undercatch, then one can expect systematic underestimation of rainfall.

Reply: Yes, you are right. The posterior probability distribution of the rainfall multipliers shows the assumption of unbiasedness is inaccurate. However, since the Bayesian inference can find the posterior probability distribution of parameters through the prior probability distribution, the prior probability distribution of the parameter does not need to be very exact. We therefore consider the assumption of unbiasedness to be acceptable. Thank you.

-line 154: it sounds like the optimal value of sigma_m was obtained "manually", but such manual calibration was criticized earlier in the introduction. Why not automate the estimation of sigma_m? Either by jointly estimating sigma_m together with the other parameters, or if possible, integrating out sigma_m from the posterior before running MCMC.

Reply: as you suggest, $\sigma$_m was jointly estimated in the revised manuscript. Thank you.

-how was sigma_e (residual standard error) in eq.6 estimated?

Reply: The calibration efficiency is highly dependent on $\sigma$_e. We artificially assigned a value to this parameter weighing accuracy and efficiency. Thank you.

-line 170: empty parentheses, missing equation?

Reply: Sorry, the storm depth multiplier "m" was missed. Thank you.

-the case study considers a very small catchment, is spatial variability of rainfall an

important source of rainfall error here (as suggested in the introduction)?

Reply: Generally speaking, the spatial variability of rainfall in small natural catchments is not obvious. But in urban areas, the local wind field is unevenly distributed caused by complex terrain. As a result, the rainfall intensity is also unevenly spatial distributed affected by the local wind field. Therefore, the spatial variability cannot be ignored even in a relatively small urban catchment.

-line 191: why are these rainfall events selected and why so few? The small number of events considered here limits robustness of any conclusions drawn from the analysis.

Reply: So far, we have only collected these data, limited by the meteorological conditions and the measurement facilities. We will supplement rainfall events data and further analyze the results in the future. Thank you.

––––––––––––––––––––

**Fig. 1.** Peak flow bias of simulations obtained through the two different calibration approaches.

[Figure]

**Fig. 2.** Total flow bias of simulations obtained through the two different calibration approaches.